# Image-Based Methods to Score Fungal Pathogen Symptom Progression and Severity in Excised *Arabidopsis* Leaves

**DOI:** 10.3390/plants10010158

**Published:** 2021-01-15

**Authors:** Mirko Pavicic, Kirk Overmyer, Attiq ur Rehman, Piet Jones, Daniel Jacobson, Kristiina Himanen

**Affiliations:** 1Oak Ridge National Laboratory, 1 Bethel Valley Rd, Oak Ridge, TN 37830, USA; pavicicvenmv@ornl.gov (M.P.); jonespc@ornl.gov (P.J.); jacobsonda@ornl.gov (D.J.); 2Department of Agricultural Sciences, Viikki Plant Science Centre, Faculty of Agriculture and Forestry, University of Helsinki, Latokartanonkaari 7, 00790 Helsinki, Finland; attiq.rehman@luke.fi; 3National Plant Phenotyping Infrastructure, HiLIFE, University of Helsinki, Latokartanonkaari 7, 00790 Helsinki, Finland; 4Organismal and Evolutionary Biology Research Program, Viikki Plant Science Centre, Faculty of Biological and Environmental Sciences, Viikinkaari 1, University of Helsinki, 00790 Helsinki, Finland; kirk.overmyer@helsinki.fi; 5Production Systems Unit, Horticulture Technologies, Natural Resources Institute (Luke), Toivonlinnantie 518, 21500 Piikkiö, Finland; 6Bredesen Center for Interdisciplinary Research and Graduate Education, University of Tennessee Knoxville, Knoxville, TN 37996, USA

**Keywords:** *Arabidopsis*, high-throughput, plant phenotyping, imaging sensors, *Botrytis*, disease symptom, chlorophyll fluorescence

## Abstract

Image-based symptom scoring of plant diseases is a powerful tool for associating disease resistance with plant genotypes. Advancements in technology have enabled new imaging and image processing strategies for statistical analysis of time-course experiments. There are several tools available for analyzing symptoms on leaves and fruits of crop plants, but only a few are available for the model plant *Arabidopsis thaliana* (Arabidopsis). Arabidopsis and the model fungus *Botrytis cinerea* (Botrytis) comprise a potent model pathosystem for the identification of signaling pathways conferring immunity against this broad host-range necrotrophic fungus. Here, we present two strategies to assess severity and symptom progression of Botrytis infection over time in Arabidopsis leaves. Thus, a pixel classification strategy using color hue values from red-green-blue (RGB) images and a random forest algorithm was used to establish necrotic, chlorotic, and healthy leaf areas. Secondly, using chlorophyll fluorescence (ChlFl) imaging, the maximum quantum yield of photosystem II (F_v_/F_m_) was determined to define diseased areas and their proportion per total leaf area. Both RGB and ChlFl imaging strategies were employed to track disease progression over time. This has provided a robust and sensitive method for detecting sensitive or resistant genetic backgrounds. A full methodological workflow, from plant culture to data analysis, is described.

## 1. Introduction

A visual assessment of plant disease symptoms is an age-old practice used for the identification of phytopathogen resistant and susceptible crops. Visual assessment is based on observations of changes in color and morphology of plants upon disease progression, rendering the method prone to variations and human error. Disease phenotyping achieved by destructive end-point assays is based on observations of pathogen colonization and quantification under microscopy, which make the analysis of disease progression cumbersome and does not allow for the following of a time series of the same event. To overcome these challenges, efforts have been made towards developing high-throughput non-invasive solutions for plant analysis and disease scoring [1,2]. These solutions are based on digitized image processing in order to provide unbiased symptom scoring. The utilization of images to evaluate diseases of plants has been practiced for over three decades [3]. Most of the available solutions use red-green-blue (RGB) imagery and chlorophyll fluorescence (ChlFl) images to quantify symptomatic areas in the whole plant or its parts [3]. As the cost of imaging technologies decreases and computing power increases, the use of both ChlFl and RGB imaging for symptom scoring has become even more common [4,5,6,7,8]. The majority of RGB and ChlFl approaches have been implemented for plants with greater agricultural importance and comparatively large organ sizes, such as tobacco, wheat, bean, and soybean [4,5,6,7,8]. The model plant *Arabidopsis thaliana* (Arabidopsis) has been used to study the identification of disease resistance genes, however, there are only a few examples found, in the literature, that use Arabidopsis with high-throughput (HTP) potential [9,10,11,12]. The grey mold, i.e., *Botrytis cinerea* (Botrytis) is the second most important necrotrophic fungus impacting agricultural production worldwide [13]. It has a host range of more than 200 crops, being especially harmful to cucumber, tomato, strawberry, raspberry, grapes, ornamental flowers, and even pine and spruce seedling production [13,14,15]. Botrytis initially tends to decrease chlorophyll content causing chlorotic lesions by yellowing/chlorosis of leaf and, being a necrotrophic fungus, it continuously produces toxic compounds that eventually cause cell death (necrosis), then, the fungus feeds on the dead tissue, hence, resulting in visible necrotic lesions [15]. Botrytis can also effectively infect Arabidopsis, rendering these two model organisms an optimal system for studying plant pathogen interactions [13,15,16]. Previously, Berger et al. [11] studied the interaction between *Pseudomonas syringae* and Arabidopsis by combining different irradiance levels and a non-biased analysis and found that using the conventional F_v_/F_m_ parameter provided a robust physiological interpretation of the plant–microbe interactions. Practically, Fm is the maximum fluorescence of a dark-adapted leaf, while F_v_ is the difference between F_m_ and the minimum fluorescence (F_o_) from a dark-adapted leaf [17,18]. Thus, the ratio F_v_/F_m_ provides an estimation of the maximum quantum efficiency of photosystem II, which usually tends to decrease during the onset of a pathogenic infection [6,10,17,18,19,20]. The benefits of chlorophyll fluorescence imaging include detecting disease in terms of F_v_/F_m_ and its progression over time, and it is also able to offer physiological information regarding overall plant health [21]. Therefore, by utilizing image-based sensors at the National Plant Phenotyping Infrastructure (NaPPI) located at the University of Helsinki, Finland, we developed a high-throughput plant disease scoring method with automated image analysis, focusing on a Botrytis-Arabidopsis model. The method was based on digital imaging which allowed simultaneous screening of several hundreds of plant organs over a time series for the analysis of disease progression. RGB and ChlFl imaging were used to develop this method, which could simultaneously screen color and physiological changes over time. Here, we provide a spatial and temporal analysis of fungal infection symptom severity during disease onset and progression.

## 2. Results

### 2.1. Botrytis Infection Symptom Screening Assay Development

To establish a sensitive disease scoring method, we used well-characterized Arabidopsis lines reported as susceptible and resistant to Botrytis [22,23]. As a susceptible control, we deployed the double knockout line *cyp79 b2/b3* (*cytochrome p450*, *family 79*, *subfamily b polypeptide 2* and *3*). This line is unable to produce both the indole alkaloid phytoalexin camalexin and indole glucosinolates, which are the most important antimicrobial compounds in Arabidopsis [22]. The mutant line *lacs2-3* (*long-chain acyl-coa synthase 2*) possesses a permeable leaf cuticle layer and is associated with a strong immunity against Botrytis, thus, it was selected as a resistant control [23]. In this way, *cyp79 b2/b3*, *lacs2-3*, and the wild type Columbia-0 (Col-0) accession represented Botrytis response controls for assessing Botrytis-induced symptom progression.

Using these three lines, an infection assay protocol was established that lasted for four weeks in total. In the first step, the seeds were sown at high density in soil, stratified in cold, and left to grow for one week. At the beginning of the second week, five seedlings per genotype were transferred to individual pots (Figure 1A). At this same time, potato carrot tomato agar (PCTA) plates were inoculated with Botrytis conidia and left to grow in darkness at room temperature (~25 °C) for two weeks in order to produce inoculum. At the beginning of the fourth week, leaves five, six, and seven from four plants per genotype (twelve in total) were excised at the base of the petiole and placed in six-well plates containing agar (Figure 1B). The leaf petioles were embedded in the agar to prevent leaf desiccation. Leaves were subsequently inoculated with a conidia suspension and the first image data were collected. The disease development was followed up to 96 h post infection utilizing both RGB and ChlFl imaging in order to record and track the appearance of symptoms (Figure 1C). Images were taken from trays containing eight six-well plates with a total of 48 leaves per tray. The workflow is summarized in Figure 1A–C.

### 2.2. RGB Image Processing Strategies

Two methods were used to measure the diseased areas in inoculated Arabidopsis leaves, namely pixel classification for RGB color hues and pixel thresholding for ChlFl imaging derived from F_v_/F_m_ values. The most obvious visible symptoms caused by Botrytis infection in Arabidopsis leaves were the tissue color changes from healthy (green) to chlorosis (yellow) and necrosis (brown). These pixel regions were quantified from the RGB images to assess the disease severity, and then followed throughout the time series in order to record the disease progression in the excised leaves (Figure 1C and Figure 2). For this purpose, an ImageJ (FIJI) plugin called “Trainable Weka Segmentation” was used [24]. This plugin allowed sampling images for different plant features to entrain a classifier to decide which category each leaf pixel belonged to (Figure 2). In this case, four categories were created, i.e., background, healthy, chlorotic, and necrotic. The categories are depicted in Figure 2 with false colors as follows: yellow (background), red (healthy), green (chlorotic), and purple (necrotic). The plugin used the pixels to train a random forest algorithm and create a classifier file that could be applied to batch images without human input. Then, this classifier was used to analyze the images of wild type, *cyp79 b2/b3*, and *lacs2-3* leaves infected with Botrytis (Figure 3A and Appendix A). This step created images with pixel counts for Categories 0, 1, 2, and 3 for the background, healthy, chlorotic, and necrotic, respectively. A grid was drawn for each leaf position in the image and the pixel values for each were stored on a spreadsheet. These results were processed in R studio where the symptomatic area was calculated by counting pixels of each category per leaf per day.

According to the color hue analysis, the wild type Col-0 plants showed some degree of all symptoms, with chlorotic being the most prevalent in the late stages of infection (Figure 3A). The hyper-susceptible *cyp79 b2/b3* mutant quickly developed necrosis, which spread to cover most of the leaf area. Little chlorosis was observed in *cyp79 b2/b3*, likely due to the rapid expansion of necrosis (Figure 3B,C). In the Botrytis-tolerant *lacs2-3* line, most of the leaf area remained classified as healthy (Figure 3C). Some necrosis and chlorosis were observed in *lacs2-3*, however, these were mostly restricted to the inoculation site and limited as compared with wild type Col-0 leaves.

### 2.3. ChlFl Image Processing

The ChlFl imaging creates monochromatic images from fluorescence coming from chlorophyll in the plant leaves. In a basic ChlFl imaging protocol, plants are dark adapted so that photosynthesis stops completely [25,26]. Plants are illuminated with red flashes that trigger a minimum fluorescence (F_o_) without starting photosynthesis and an image is acquired. Another image is recorded with a saturating pulse of light that induces a maximum peak of fluorescence (F_m_). The F_o_ pixel values are subtracted from F_m_ pixel values to create an image termed F_v_ or variable fluorescence. The F_v_ pixel values divided by F_m_ pixel values give F_v_/F_m_ or the maximum quantum yield of the photosystem II if all reaction centers are open. The F_v_/F_m_ in healthy plants has a value of ~0.83 and decays rapidly when plants are stressed [26]. In summary, the image output of a basic ChlFl imaging protocol is F_o_, F_m_, F_v_, and F_v_/F_m_ images, of which the decay of F_v_/F_m_ can be used to record the disease progression.

The calculation of F_v_/F_m_ results in a very noisy image background (Figure 2). To identify only leaf pixels in the F_v_/F_m_ images, a background masking was required. The F_o_ images, where the leaves were clearly distinguishable from the background, allowed for the creation of the masks (Figure 2).

The F_v_/F_m_ values at 72 h post infection of wild type, *cyp79 b2/b3,* and *lacs2-3* lines, are shown in Figure 4A. Wild type Col-0 plants showed strong F_v_/F_m_ decay at the inoculation site, which expanded to some extent to the neighboring pixels, while a good portion of the leaves remained close to the healthy value of 0.8. In the susceptible *cyp79 b2/b3* mutant, the F_v_/F_m_ values decayed strongly, covering nearly the whole leaf with only a small portion of some leaves remaining healthy. Conversely, the *lacs2-3* F_v_/F_m_ pixel values remained healthy, decaying slightly only at the inoculation point. A density plot of pixel value distribution of F_v_/F_m_ images showed that most of the leaf pixels stayed above 0.75 in the *lacs2-3* leaves (Figure 4B). Using this information, we defined the pixel value threshold of 0.75 to be the cutoff for symptomatic pixels; values below 0.75 were considered to be symptomatic. In this way, two parameters were used to track Botrytis infection in F_v_/F_m_ images, namely the increase in size of the symptomatic area, i.e., a count of pixels with value below 0.75, and the severity of the infection, calculated as total disease pixel area per whole leaf area.

Figure 4C shows the progression of the disease symptomatic pixel count from ChlFl and Figure 4D shows the severity of the infection by F_v_/F_m_ decay in the three control lines. The line *cyp79 b2/b3* showed a larger symptomatic pixel count and stronger infection severity than wild type plants. The opposite effect was observed in *lacs2-3* leaves, thus validating the method.

It should be noted that these datasets presented a nonlinear behavior and an increased dispersion along the time course. The RGB data and symptomatic F_v_/F_m_ area severity are proportional data with values between zero and one, while the symptomatic pixel area is count data. Furthermore, these are not independent measurements since we measured the same individuals several times, therefore, the measurements are not independent from each other and this should be included in the model. To cope with these data features, a generalized additive mixed model (GAMM) using Poisson probability distribution was used for symptomatic pixel counts and a beta probability distribution was used for the other parameters. The differences among mutant lines and wild type were established by differences in the goodness of fit using Akaike information criterion (AIC) of a GAMM with and without the genotype term.

The AIC difference establishes that one model has a better fit than another, where the lower the AIC, the better the fit. For example, the healthy pixel proportion of Col-0 and *lacs2-3* are shown in Figure 5. In this case, a base model was fit without indicating their genotype (Figure 5A). A second model (full model) was fit to the same data, and indicated that there were two genotypes, i.e., Col-0 and *lacs2-3* (Figure 5B). The AIC values of both models were calculated, and their AIC difference was nine units suggesting that the model fit in Figure 5B was the best model, and therefore leaf genotype has a significant impact on healthy pixel proportion decay (Figure 5). For statistical inference, AIC difference ≤2 show support for the model without the genotype term, 4≤ AIC difference ≤7 show considerably less support, and AIC difference >10 show no support, and therefore the genotype term should be included in the model [27].

This process was repeated for each parameter for both *lacs2-3* and *cyp79 b2/b3*. For most of the parameters analyzed, *lacs2-3* and *cyp79 b2/b3* were significantly different from Col-0 leaves (Table 1). Only *cyp79 b2/b3* healthy pixel proportion and *lacs2-3* necrotic pixel proportion were not significant. Nonetheless, the symptom trend showed that *cyp79 b2/b3* had a lower proportion of healthy pixels and *lacs2-3* had a lower proportion of necrotic pixels than Col-0 (Appendix A).

## 3. Discussion

Botrytis, also known as grey mold, is the second most important fungal pathogen in agriculture worldwide with a wide host range of more than 200 crops [13,14,15]. There are several transcriptome studies available and many genes have been identified to be activated in plants upon contact with Botrytis, but it remains unclear if they are responsible for plant immunity and/or posterior disease development. Utilization of genetically engineered model plants has significantly advanced our understanding of plant pathogenesis and unraveled the role of many genes involved in diseases. In contrast to biotrophic pathogens, which have simple gene-for-gene resistance involving a single *Resistance* (*R*) gene in the host, resistance against necrotrophs such as Botrytis, is complex, quantitative, and polygenic. For this reason, despite recent advances, our understanding of immune signaling mechanisms against necrotrophs lags behind that of biotrophs [22,28,29]. Thus, there is a need for further studies into the genetics of plant immunity against Botrytis. The ability of Botrytis to infect Arabidopsis offers an excellent infection system to explore gene function, due to the enormous amount of information and genetic tools available for Arabidopsis. Genes identified through this system could be used for breeding disease resistance in crop plants [30]. Pathogen infection phenotyping is, however, currently time-consuming, and subjective or end-point assays make the analysis of disease progression non-viable [3]. To overcome this, many digital image-based tools have been developed with automation potential, providing the opportunity to develop an unbiased and high-throughput analysis.

Pathogens induce many physiological changes in plants that can be detected by image-based analysis [11]. Stressed plants exhibit visible symptoms such as chlorosis and necrosis at the infection sites. An analysis of these symptoms is traditionally based on a visual assessment of the color of the infection site and measurements of the size of the infection spot. Both these measurements are laborious and can be biased by human error. Automated image analysis that is based on pixel classification provides a novel alternative to these manual assays. In addition to visible responses, plants have many invisible changes that occur during infections. These changes can be detected by chlorophyll fluorescence measurements that represent the photosynthetic activity in the plant. Thus, these measurements can detect physiological responses early, for example, as early as 6 h after bacterial inoculation [11]. Furthermore, early, and late responses can be differentiated by specific chlorophyll fluorescence signatures.

In this study, RGB and ChlFl cameras were used to record Botrytis disease progression in detached leaves of Arabidopsis. The sensors are mounted on a conveyor belt system that allows automatic tray delivery and imaging (Figure 1A). The RGB images record the reflected light of red, green, and blue colors that are stored as three channels in the image. Then, this information is used to create a full color image using the different levels of red, green, and blue in the picture. In this way, each pixel in the image stores information for these three values (red, green, and blue) that act as coordinates for any image software to compose an image on the screen. This information can also be used to classify each pixel into different groups. There are several classification algorithms available such as k-means, k-nearest neighbor, logistic regression, decision trees, random forests, etc. In this study, a random forest algorithm was trained to create a classification model. Random forests are based on collections of decision trees, where many trees vote on which category a pixel should be classified. This has the advantage of correcting the tendency of individual decision trees to overfit the training dataset. Every classification method has some degree of inaccuracy where some pixels are misclassified, especially those values close to the category borders. For instance, in Appendix A, some chlorotic pixels were identified by manual segmentation, however, the trained model was unable to recognize those, showing room for further training and improvement of the model. Therefore, it is important to use proper controls and expert evaluation that allows validation of the classifications. Classification models ease the post processing data extraction, since it can be automatically applied to batches of images. The RGB analysis in this study utilizes completely free tools, and therefore can be applied to any picture of infected leaves taken in a systematic manner in any laboratory.

ChlFl imaging is classified as active imaging since it requires its own light source. ChlFl imaging is more sensitive than RGB imaging and can detect symptoms earlier than RGB imaging. Due to the monochromatic nature of ChlFl images, the symptoms are scored by averaging all pixel values in the leaf to estimate the infection severity or by classifying them as symptomatic or not symptomatic using pixel thresholding to estimate the symptomatic area of the leaf. Because ChlFl imaging depends on chlorophyll, it is difficult to detect signals when the infection is very advanced and little chlorophyll remains in the leaf. This event can create an underestimation of the diseased area. Therefore, ChlFl is a sensitive technique that should be used with care with aggressive pathogens.

Data can sometimes be analyzed by linear regression with a Gaussian probability distribution. In order to properly apply this technique, the data must fulfil the following assumptions: (i) measurements should be normally distributed, (ii) each measurement should be independent, and (iii) the variance of the measurements should be homogeneous across treatments (homoscedasticity). In this study, the data violated all these assumptions making Gaussian linear regression inappropriate for these datasets. For instance, symptomatic pixel counts cannot take values below zero and symptomatic pixel proportions and F_v_/F_m_ can only take values between zero and one. To solve this problem, a Poisson probability distribution was used for symptomatic pixel counts and a beta probability distribution was used for proportional data. All parameters analyzed exhibited nonlinear patterns and a tendency for over dispersion along the time course, violating the homoscedasticity assumption of Gaussian models. Thus, GAM models were used to account for nonlinear patterns and overdispersion. Finally, to deal with the dependency created by repeated measurements, each leaf was labeled with an identification (id) that was included as random effect in the GAM.

## 4. Conclusions

In this study, we have proposed a complete pipeline for the analysis of Botrytis symptom progression in Arabidopsis. This pipeline is based on RGB and ChlFl images, where the full image and data processing are scripted, opening the possibility to use this method in HTP applications. The RGB image analysis was performed using freeware and could be implemented in any lab with any standard RGB camera. Here, we also show that phenotyping data may violate many modeling assumptions, and therefore it is of utmost importance to use the proper statistical tools to analyze different data types, otherwise, incorrect inferences may be drawn. In addition, many of the methodological steps regarding sample collection and inoculation still requires manual work, in turn affecting the throughput of the study and can serve as a bottleneck for genetic screening of large populations. Nonetheless, this study could serve as a useful tool for screening Arabidopsis germplasm in various infection assays and could be extrapolated to other plant species with modifications in our open-source analysis pipeline using freely available tools such as FIJI and R.

## 5. Materials and Methods

### 5.1. Plant Material and Growth Conditions

To develop a disease scoring method, the cuticle permeable and Botrytis-resistant *long-chain acyl-coa synthase 2* (*lacs2-3*) mutant and camalexin and indole glucosinolate deficient and Botrytis-susceptible *cytochrome p450 79-b2* and *-b3* (*cyp79 b2/b3*) double mutant were used as controls (Bessire et al., 2007 and Buxdorf et al., 2013). The wild type Columbia-0 (Col-0) accession was used as a reference control. Seeds were sown in water saturated vermiculite-peat (Type B2 peat, Kekkilä, Vantaa, Finland, www.kekkila.fi) substrate (1:1 ratio) in 8 × 8 cm pots at high density (~100 seeds per pot). Then, pots were put in mini greenhouses and stratified at 4 °C for 72 h, in darkness, to ensure even germination. Mini greenhouses were placed in a growth room (PhytoScope, PSI, Drasov, Czech Republic) and kept covered to maintain humidity during germination. On day four, the lid was removed, and seedlings were manually thinned using forceps leaving 30 seedlings per pot. Growth conditions in the Arabidopsis growth chamber were 12 h light/12 h darkness and 22 °C and 60% relative air humidity. White LED was used as light source with a photosynthetically active radiation (PAR) of 130 µmol·m^−2^·s^−1^, controlled using an LI-190R Quantum Sensor coupled to an LI-250A light meter (LI-COR, Bad Homburg, Germany, www.licor.com).

### 5.2. Botrytis Cinerea Culture Conditions

Botrytis strain B05.10 was cultured on potato carrot tomato agar (PCTA) medium containing potato carrot tomato extract (see Appendix A for details), 1% dextrose (Sigma-Aldrich, Munich, Germany, www.sigmaaldrich.com), 0.15% yeast extract (Biokar Diagnostics, Allonne, France), and 0.7% agar (Sigma-Aldrich). A detailed description of the PCTA medium preparation is provided in Appendix B. Botrytis inoculum was cultured for two weeks in PCTA plates in darkness at room temperature (~23 °C). Botrytis conidia were collected using 2/3 strength potato dextrose broth (Sigma-Aldrich) and filtered using miracloth (Merk Millipore, Darmstadt, Germany, www.merckmillipore.com) to remove mycelia. Conidia concentration was adjusted to 1 × 10^6^/mL using a Fuchs-Rosenthal counting chamber (Assistent, Germany, www.hecht-assistent.de).

### 5.3. Dissected Leaf Infection Assay

For leaf infections, one-week-old Arabidopsis seedlings were transferred to individual pots (6 cm^2^) and allowed to grow for two weeks more (Figure 1A). Whenever possible, leaves five, six, and seven were harvested from three-week-old plants and placed in six-well plates (Sarstedt, Nümbrecht, Germany, www.sarstedt.com) containing 0.7% agar gel (Duchefa Biochemie, Haarlem, The Netherlands, www.duchefa-biochemie.com) and distributed in trays holding eight plates each (Figure 1B). Twelve leaves per line from four individual plants were inoculated with 10 µL of the conidia suspension. The infection development was recorded daily for a period of 96 h post infection.

### 5.4. Phenotyping Infrastructure

The disease scoring method was developed utilizing the imaging sensors available at the NaPPI facility at the University of Helsinki (https://www.helsinki.fi/en/infrastructures/national-plant-phenotyping). This unit has a controlled environment FytoScope Walk-In chamber with a PlantScreen™ Compact System and 12 LED-illuminated shelves as an integrated phenotyping platform (Photon Systems Instruments, PSI, Drasov, Czech Republic, www.psi.cz). The PlantScreen™ Compact System can hold up to 18 trays and the growth shelves another 36 trays of 20 plants each, making a potential total of 2592 leaves per experiment. Within this system, trays were transported on conveyor belts between the light-isolated visible RGB imaging cabinet, chlorophyll fluorescence (ChlFl) imaging cabinet, the weighing and watering station, and the dark/light acclimation chamber with regulated photosynthetically active radiation (PAR) from 0–700 μmol m^−2^ s^−1^. The ChlFl measurements were acquired using an enhanced version of the FluorCam FC-800MF pulse amplitude modulated (PAM) chlorophyll fluorometer (PSI, Czech Republic). The ChlFl imaging unit has been described by Awlia et al. (2016) and features a 1/2” monochromatic sensor of 720 × 560 pixels resolution and a lens type Lensagon CY0314. The ChlFl illumination panel (FluorCam SN-FC800-195) has a pulse-modulated short duration red-orange flashes (620 nm), a red-orange actinic light (620 nm) with maximum photosynthetic photon flux density (PPFD) of 300 μmol m^−2^ s^−1^, a cool white actinic light with maximum PPDF of 500 μmol m^−2^ s^−^, and a saturating light pulse with a maximum PPDF of 3000 μmol m^−2^ s^−1^. The RGB imaging unit is equipped with three RGB cameras (one top and two side views) mounted on robotic arms, each supplemented with an LED-based light source to ensure homogeneous illumination of the imaged object. The RGB images (resolution 2560 × 1920 pixels) of individual trays were captured using the GigE uEye UI-5580SEC/M 5 Mpx Camera (IDS, Obersulm, Germany) with SV-0814H lens to assess plant growth and morphological traits. Light conditions, plant position, and camera settings were fixed throughout the experiments.

### 5.5. Imaging Recording and Processing

Following inoculation, chlorophyll fluorescence and RGB imaging was performed daily, for five days. The RGB images were processed using Morpho Analysis software for background removal (version 1.0.7, PSI). The FIJI software was subsequently used with the Weka segmentation plugin to train a random forest algorithm for classification of leaf pixels into the following four classes: healthy, chlorotic, necrotic, and background. Thus, using the GUI, pixels representing each class were sampled manually to train the model. The training features selected to create the model were Gaussian blur, Sobel filter, Hessian, difference of Gaussian, and membrane projections. Once trained, the classifier was applied to all images in batch mode using FIJI scripting tool according to the ImageJ wiki page “Scripting the Trainable Weka Segmentation” (https://imagej.github.io/Scripting_the_Trainable_Weka_Segmentation). Afterwards, a grid was drawn with squares matching each leaf position on the classified images and the pixel information was extracted using the FIJI function “save XY coordinates”. Pixel values were further processed using R studio (www.rstudio.com), as described below.

The ChlFl imaging-based detection of the infection site was done based on reduction of the chlorophyll fluorescence emitted from the infected leaf. Image capture was done using the principal F_v_/F_m_ imaging protocol of a FluorCam system (PSI) that generated parameter images of minimum fluorescence (F_o_) and maximum fluorescence (F_m_) yields [26,31]. To score the infection severity, a common plant stress indicator, i.e., the quantum yield of photosystem II (F_v_/F_m_), was utilized [11,32]. The ChlFl raw images of .fimg format were collected for five days and stored in the central database. These images were processed to extract pixel information using the FIJI software [33,34]. Due to the high number of images produced, image processing was scripted, and run in batch mode, in FIJI. The script steps were the following:Open F_o_ images;Select background pixels by pixel thresholding;Convert to binary images and save them to be used as masks;Open masks and select background pixels;Open F_v_/F_m_ images;Transfer background selection from masks to F_v_/F_m_ images;Set F_v_/F_m_ background pixels value to −100;Draw a rectangle for each plant position;Save XY coordinates to export each pixel value within the rectangle as comma separated values (CSV) file.

The pixel threshold was arbitrarily selected for each image in order to select as much leaf area as possible. In advanced stages of the infection, pixel values in the infected area were too low to be selected by normal thresholding and the masks corresponding to these time points were manually modified to obtain a better estimation of the symptomatic area.

The output of this script generated one CSV file per leaf per time point that were imported and further analyzed using R studio software.

### 5.6. Data Processing in R Studio

Due to fluctuation in the leaf size, the pixel count in each category of RGB images was normalized against the total leaf size and results were displayed as stacked color plots. For ChlFl results, the CSV files were merged into a single file and leaf genotype information was added to this dataset. Background pixels were filtered out by removing all values equal to −100. The infection size was calculated by counting the number of pixels with values below 0.75 and the infection severity was calculated by averaging all leaf pixels. This data processing and analysis was done using the Tidyverse R group packages (www.tidyverse.org/packages). Statistical analysis was done using R package mgcv and generalized additive models with different probability distributions depending on the data type [35]. Then, Akaike information criterion (AIC) was calculated for model inference [27]. All R and ImageJ script generated to process are available at https://github.com/mipavici/MDPI_leaf_infection.

## Figures and Tables

**Figure 1 plants-10-00158-f001:**
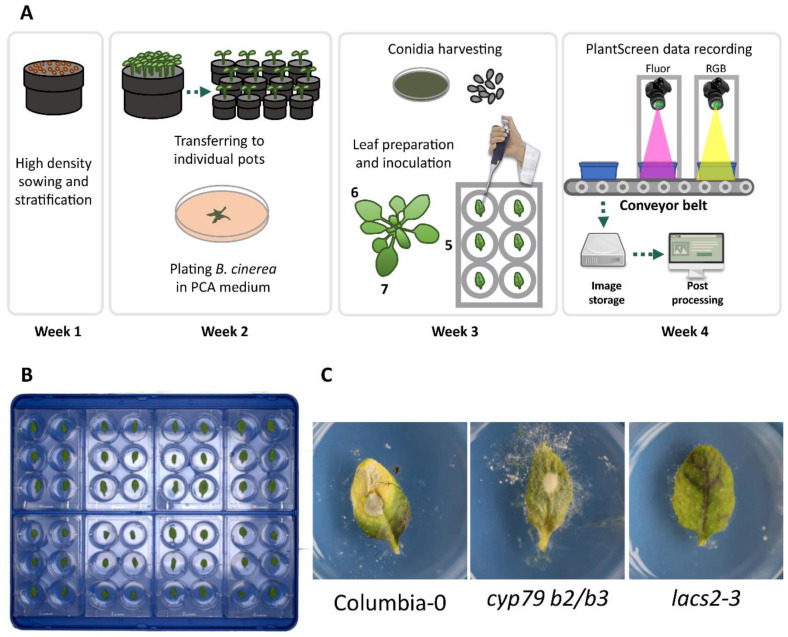
Arabidopsis leaf infection workflow. (**A**) The workflow of the infection assay used in this study starting from high density sowing of Arabidopsis seeds, potting of seedlings into individual pots, Botrytis conidia culture, inoculation of excised Arabidopsis leaves, imaging by red-green-blue (RGB) and chlorophyll fluorescence (ChlFl) sensors, online image storage and post processing, and data analysis; (**B**) Eight six-well plates arranged on the imaging tray; (**C**) Control lines Col-0, *cyp79 b2/b3,* and *lacs2-3* used in this study, after 96 h post inoculation. PCTA medium, potato carrot tomato agar medium; ChlFl, chlorophyll fluorescence imaging; RGB imaging, red-green-blue imaging; Col-0, wild type Columbia-0 accession; *cyp79 b2/b3*, the Botrytis-susceptible *cytochrome p450 79-b2* and *-b3* (*cyp79 b2/b3*) double mutant; *lacs2-3*, the Botrytis-resistant *long-chain acyl-coa synthase2* (*lacs2-3*) mutant.

**Figure 2 plants-10-00158-f002:**
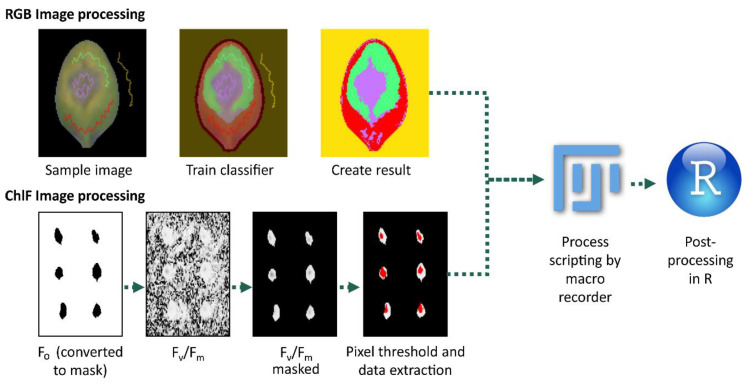
Image processing workflow for RGB (red-green-blue) and ChlFl (chlorophyll fluorescence) images. For RGB images, four categories were depicted, i.e., background, healthy, chlorotic, and necrotic, and are indicated by yellow, red, green, and purple colors, respectively. For ChlFl images, an F_o_ masked image was used to detect the leaf F_v_/F_m_ to record the photosynthetic capacity on the leaf and the two together to assign pixel thresholds for disease. Both image derived datasets were processed in FIJI and analyzed by R scripts. F_o_, minimum fluorescence; F_v_, variable fluorescence; F_m_, maximum fluorescence; F_v_/F_m_, maximum quantum yield of the photosystem II.

**Figure 3 plants-10-00158-f003:**
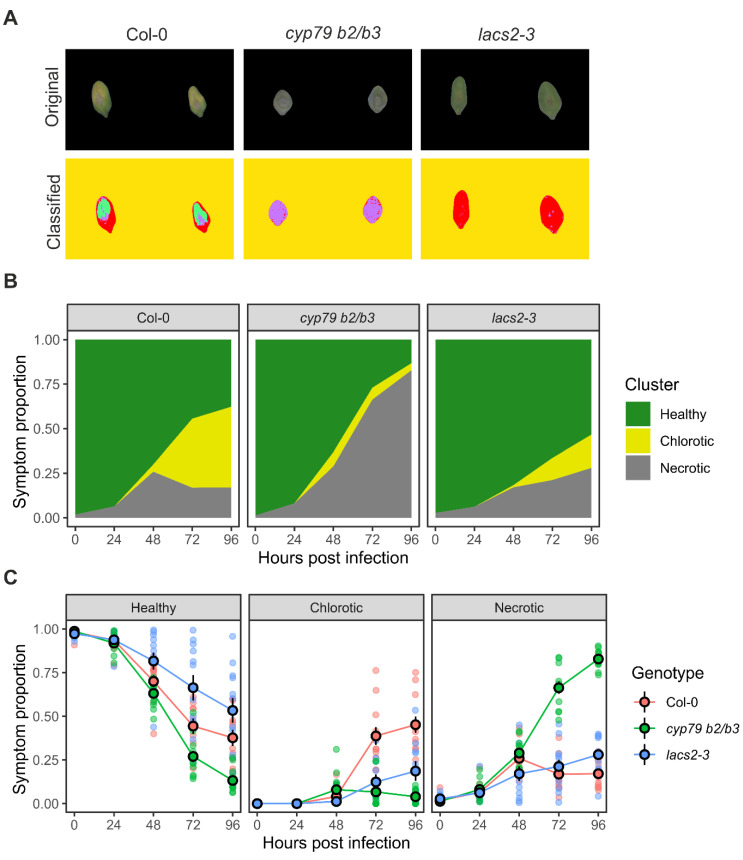
Validation of the RGB pixel classification strategy for Botrytis disease scoring. (**A**) Images of original RGB images (above) and pixel classification results with color codes (below); (**B**) Stacked color hue plots of the diseased area progression for the symptom categories healthy, chlorotic, and necrotic; (**C**) Disease progression area for healthy, chlorotic, and necrotic categories. Circles, mean; error bars, standard error of the mean; Col-0, wild type Columbia-0 accession; *cyp79 b2/b3*, the Botrytis-susceptible *cytochrome p450 79-b2* and *-b3* double mutant; *lacs2-3*, the Botrytis-resistant *long-chain acyl-coa synthase2* mutant.

**Figure 4 plants-10-00158-f004:**
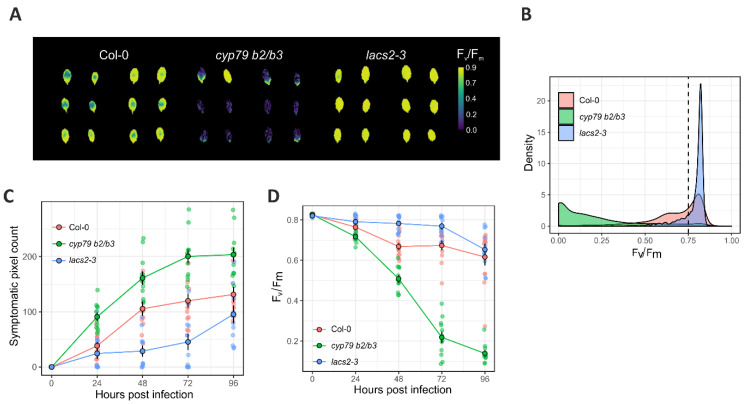
Chlorophyll fluorescence threshold pixel count and mean pixel value strategies for Botrytis disease scoring. (**A**) False color F_v_/F_m_ image. Yellow pixels represent healthy leaf areas, while green and blue represent symptomatic areas; (**B**) Density plot for the distributions of pixel intensities for wild type Col-0, *cyp79 b2/b3,* and *lacs2-3*, at 72 h post infection, with an arbitrary pixel threshold (≤0.75) to consider a pixel as symptomatic (dashed line); (**C**) Symptomatic pixel count and (**D**) disease severity (mean pixel value over the leaf) differences among Col-0, *cyp79 b2/b3,* and *lacs2-3*. Circles represent the mean, and the error bars the standard deviation from the mean. Transparent points in the background represent the actual individual leaves measured with a soft jittering to prevent point overlapping. Col-0, wild type Columbia-0 accession; *cyp79 b2/b3*, the Botrytis-susceptible *cytochrome p450 79-b2* and *-b3* double mutant; *lacs2-3*, the Botrytis-resistant *long-chain acyl-coa synthase2* mutant.

**Figure 5 plants-10-00158-f005:**
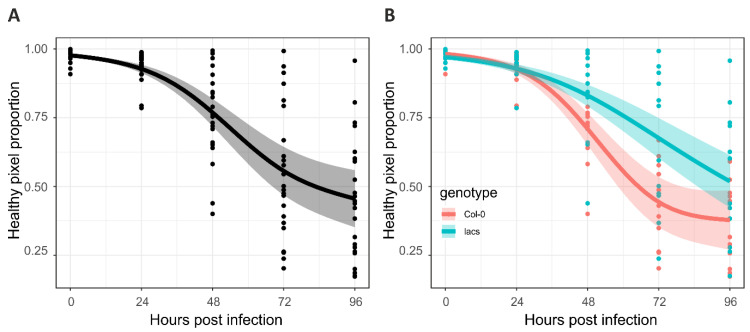
Modeling strategy for inference. (**A**) Base model without indicating leaves genotype; (**B**) Full model including leaves genotype term. Col-0, wild type Columbia-0 accession; *lacs2-3*, the Botrytis-resistant *long-chain acyl-coa synthase2* (*lacs2-3*) mutant.

**Table 1 plants-10-00158-t001:** Statistical output of the relative likelihood test performed in this study.

Line	Imaging	Parameter	AIC Base Model	AIC Full Model	AIC Diff.
*cyp79 b2/b3*	RGB	Healthy	−410.2	−412.5	2.2
		Chlorotic	−1302.1	−1325.4	23.3
		Necrotic	−322.8	−364.7	41.9
	ChlFl	Pixel count	1492.0	1305.5	−186.5
		Severity	−341.8	−382.9	41.1
*lacs2-3*	RGB	Healthy	−373.3	−383.0	9.7
		Chlorotic	−1449.7	−1448.5	1.3
		Necrotic	−334.3	−339.6	5.3
	ChlFl	Pixel count	2174.9	1799.3	−375.6
		Severity	−413.9	−458.5	44.5

Col-0, wild type Columbia-0 accession; *cyp79 b2/b3*, the Botrytis-susceptible *cytochrome p450 79-b2* and *-b3* double mutant; *lacs2-3*, the Botrytis-resistant *long-chain acyl-coa synthase2* mutant; RGB, red-green-blue; ChlFl, chlorophyll fluorescence; AIC, Akaike information criterion; Diff, difference.

## Data Availability

The data presented in this study are openly available at https://github.com/mipavici/MDPI_leaf_infection.

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
