# Peer review of "Image-Based Methods to Score Fungal Pathogen Symptom Progression and Severity in Excised Arabidopsis Leaves"

_plants, 2021, doi:10.3390/plants10010158_

Round 1
Reviewer 1 Report
- Pipeline codes as well as machine learning model weights file should be deposited to a repository or as a supplementary to assure reproducibility.
- State what kind of image features have you used to train the classifier.
- The model performance (e.g. IoU, ROC) of the random forest classifier model is not present in both training dataset (and test dataset, which I assume possibly not prepared). The authors must present in any form to assure the model accuracy before presenting any biological results. Systematically evaluating many test images is preferable, however can be alternatively done by the plot result function of weka. The least preferable but acceptable is aligning several representative images from genotypes in different days with machine inferred label and hand annotation, along with original image somewhere in supplementary.
Author Response
We appreciate the reviewer's time and suggestions for the improvement of our study. We have incorporated all the requested changes in the revised version of the manuscript and supplemented the necessary information in respective sections. A new supplementary figures was added to the manuscript as well.
Reviewer 1 comment: Pipeline codes as well as machine learning model weights file should be deposited to a repository or as a supplementary to assure reproducibility.
Author response: The complete set of codes, models and necessary files are uploaded on Github at https://github.com/mipavici/MDPI_leaf_infection. The link is also provided in the method section and a tutorial is being prepared and will be uploaded to the Github repository. To ensure maximum reproducibility the images used in this study are also available at the repository. This information has been now supplemented in the revised method section (page 12, line 429).
Reviewer 1 comment: State what kind of image features have you used to train the classifier.
Author response: We used training features such as gaussian blur, sobel filter, hessian, difference of gaussian and membrane projections to create the classifier models. This information has been now supplemented in the revised method section (page 11, line 386).
Reviewer comment: The model performance (e.g. IoU, ROC) of the random forest classifier model is not present in both training dataset (and test dataset, which I assume possibly not prepared). The authors must present in any form to assure the model accuracy before presenting any biological results. Systematically evaluating many test images is preferable, however can be alternatively done by the plot result function of weka. The least preferable but acceptable is aligning several representative images from genotypes in different days with machine inferred label and hand annotation, along with original image somewhere in supplementary.
Author response: We created a new supplementary file where manually segmented leaves are compared with those segmented by the trained model. The figure shows examples of all three genotypes for each time point analyzed. We added a short discussion about this new image in the discussion section (page 9, lines 280 - 283).
We thank the reviewer 1 for the constructive feedback on our work.
Best regards,
Authors
Reviewer 2 Report
The manuscript provides a methodology for image-based detection of lesions generated by inoculating Arabidopsis leaves (wild-type and mutants) with B. cinerea conidia. The effects post-inoculation on leaves are followed over time to derive differential sensitivity to this fungus.
The manuscript is clear and well designed, with clear illustrations and data analysis.
There are a few points that could be improved for the benefit of a borader readership.
- Somehow it may not be obvious to many readers what is a chlorotic vs. necrotic lesion. It is likely useful to indicate this point clearly when lesion types are defined.
- The authors use a freely available machine learning algorithm implemented in the Image J (Fiji) software. Because Image J and this algorithm are freeware and publicly available, it should be stressed that they are potentially usable by a vast numbers of researchers. However, it would be useful to include more information about the 'Weka Segmentation Tool'.
- The methodology is described by high-throughput. However, some steps, such as leaf explant and plating are likely to be time-consuming. It would be useful to include some considerations about the encountered bottlenecks towards increasing throughput for genetic screening (i.e., for association mapping with hundreds of genotypes).
Author Response
We cordially thank the reviewer 2 for appreciating the merits of our study and giving constructive feedback to improve our work. We hereby address all the comments by incorporating necessary changes in the text and by providing additional information as required.
Reviewer 2 comment: Somehow it may not be obvious to many readers what is a chlorotic vs. necrotic lesion. It is likely useful to indicate this point clearly when lesion types are defined.
Author response: Development and distinctness of chlorotic and necrotic lesions caused by B. cinerea has been added in the Introduction section (page 2, lines 59 - 62) with proper citation.
Reviewer 2 comment: The authors use a freely available machine learning algorithm implemented in the Image J (Fiji) software. Because Image J and this algorithm are freeware and publicly available, it should be stressed that they are potentially usable by a vast number of researchers. However, it would be useful to include more information about the 'Weka Segmentation Tool'.
Author response: Information related to ‘Weka Segmentation Tool’ has been supplemented and also a Github link the plugin details is provided in the method section (page 12, line 429). Our pipeline is also uploaded on Github and is made available to other researchers by accessing at https://github.com/mipavici/MDPI_leaf_infection
Reviewer 2 comment: The methodology is described by high-throughput. However, some steps, such as leaf explant and plating are likely to be time-consuming. It would be useful to include some considerations about the encountered bottlenecks towards increasing throughput for genetic screening (i.e., for association mapping with hundreds of genotypes).
Author response: Many thanks for pointing this out and we absolutely agree on the limitations of the work. A self-reflection about the potential bottlenecks related to methodology has been acknowledged in the conclusions section of the revised version of the manuscript (page 9, lines 316 - 321).
We again appreciate the reviewer's time to go through our work and their kind recommendations.
With best wishes,
Authors